# Biological effects related to exposure to polychlorinated biphenyl (PCB) and decabromodiphenyl ether (BDE-209) on cats

**Kraisiri Khidkhan**[1,2]**, Hazuki Mizukawa**[2,3]**, Yoshinori Ikenaka**[2,4,5]**, Shouta M. M. Nakayama**[2]**, Kei Nomiyama**[6]**, Nozomu Yokoyama**[2]**, Osamu Ichii**[2,7]**, Mitsuyoshi Takiguchi**[2]**, Shinsuke Tanabe**[6]**, Mayumi Ishizuka**[2]*

1 Department of Pharmacology, Faculty of Veterinary Medicine, Kasetsart University, Bangkok, Thailand, 2 Faculty of Veterinary Medicine, Hokkaido University, Sapporo, Hokkaido, Japan, 3 Department of Science and Technology for Biological Resources and Environment, Graduate School of Agriculture, Ehime University, Matsuyama, Ehime, Japan, 4 Water Research Group, Unit for Environmental Sciences and Management, North-West University, Potchefstroom, South Africa, 5 One Health Research Center, Hokkaido University, Sapporo, Hokkaido, Japan, 6 Center for Marine Environmental Studies (CMES), Ehime University, Matsuyama, Ehime, Japan, 7 Laboratory of Agrobiomedical Science, Faculty of Agriculture, Hokkaido University, Sapporo, Hokkaido, Japan

* ishizum@vetmed.hokkudai.ac.jp

**Data Availability Statement:** Data are within the paper and its Supporting information files.

## Abstract

As an animal familiar to humans, cats are considered to be sensitive to chemicals; cats may be exposed to polychlorinated biphenyls (PCBs) and decabromodiphenyl ether (BDE-209) from indoor dust, household products, and common pet food, leading to adverse endocrine effects, such as thyroid hormone dysfunction. To elucidate the general biological effects resulting from exposure of cats to PCBs and PBDEs, cats were treated with a single i.p. dose of a principal mixture of 12 PCBs and observed for a short-term period. Results revealed that the testis weight, serum albumin, and total protein of the treated group decrease statistically in comparison with those in the control group. The negative correlations suggested that the decrease in the total protein and albumin levels may be disturbed by 4′OH-CB18, 3′OH-CB28 and 3OH-CB101. Meanwhile, the serum albumin level and relative brain weight decreased significantly for cats subjected to 1-year continuous oral administration of BDE-209 in comparison to those of control cats. In addition, the subcutaneous fat as well as serum high-density lipoprotein (HDL) and triglycerides (TG) levels increased in cats treated with BDE-209 and down-regulation of stearoyl-CoA desaturase mRNA expression in the liver occurred. These results suggested that chronic BDE-209 treatment may restrain lipolysis in the liver, which is associated with lipogenesis in the subcutaneous fat. Evidence of liver and kidney cell damage was not observed as there was no significant difference in the liver enzymes, blood urea nitrogen and creatinine levels between the two groups of both experiments. To the best of our knowledge, this is the first study that provides information on the biochemical effects of organohalogen compounds in cats. Further investigations on risk assessment and other potential health effects of PCBs and PBDEs on the reproductive system, brain, and lipid metabolism in cats are required.

**Funding:** This work was financially supported by Grants-in-Aid for Scientific Research (B) (No. 16H02989), Challenging Exploratory Research (No. 17K20038) and Scientific Research (A) (No. 21H04919) from the Japan Society for the Promotion of Science and partly supported by the Ministry of Education, Culture, Sports, Science, and Technology (MEXT), Japan, to a project on Joint Usage/Research Center Leading Academia in Marine and Environmental Research (LaMer), Ehime University. This research was also supported by Sousei Tokutei Research of Hokkaido University. The funders had no role in study design, data collection and analysis, decision to publish, or preparation of the manuscript. There was no additional external funding received for this study.

**Competing interests:** The authors have declared that no competing interests exist.

## Introduction

Polychlorinated biphenyls (PCBs) and polybrominated diphenyl ethers (PBDEs) are two classes of manmade organohalogen compounds (OHCs) that are commonly used as pigments and flame retardant additives in the manufacture of industrial materials and several indoor household products such as furniture, electrical equipment, plastic, and textiles [1, 2]. Although PCBs and PBDEs are beneficial for the fire protection of properties and lives, these compounds are ubiquitous contaminants; several studies on the accumulation levels of PCBs and PBDEs in humans and surrounding environment are reported worldwide [3–5]. The main indoor routes of PCB and PBDE include ingestion and dermal absorption of house dust [6–8]. These two chemicals are among the most toxic persistent substances; therefore, the production of PCBs and PBDEs has been banned by the Stockholm Convention [9]. As their structures are similar to those of endocrine hormones such as thyroxin, PCBs and PBDEs are known to cause liver and endocrine system dysfunction, thyroid toxicity, developmental neurotoxicity, gonadal dysfunction and possibly cancer in human, livestock and wildlife [4, 10–12].

Domestic cats are good friends to humans; usually, cats share living spaces with their owners [13]. In veterinary and toxicology studies, cats have been regarded as a good sentinel species for human exposure to indoor pollutions, especially the contamination of persistent organic chemicals including PCBs and PBDEs [14–19]. Among PBDE congeners, BDE-209 was detected in the highest proportion in human and pet sera [20–24]. Several biomonitoring studies reported that feline hyperthyroidism is associated with PCB and PBDE levels in the cat serum; these cats are likely to be subjected to chronic exposure to organohalogen substances via contaminated foods and indoor house dust [21, 25–28]. On the other hand, a recent *in vivo* study (single-dose exposure) has reported that PCBs could not change levels of thyroid hormones (including free T4, total T4, free T3, total T3, and TSH) in the serum of cats [29], but it could induce the liver mRNA expression of cytochrome P450 family 1 genes (*viz.*, CYP1A1, CYP1A2 and CYP1B1) [30]; these genes are involved in an aryl hydrocarbon-receptor-mediated signaling pathway that contributes to a broad range of physiological roles, including immune function, organ development, reproduction, and steroid signaling modulation [31]. However, limited information on the effects of PCB or PBDE on cats in terms of physiological and biochemical changes is available thus far.

Thus, in this study, the acute effects of PCB and chronic effects of PBDE, especially BDE-209, were investigated by (1) determination of clinical signs, body weights, organ weights, biochemical parameters, and steroid hormone levels of the serum of cats subjected to single-dose exposure of PCB; and (2) examination of clinical signs, body weight, organ weight, serum biochemical parameters, serum PBDE, and its hydroxylated metabolites, as well as gene expressions in the liver of cats subjected to long-term treatment of BED-209.

## Materials and methods

### Animal and sample collections

All experiments were approved for ethical compliance by the Hokkaido University (approval numbers 14–0054 and 14015) and were performed in accordance with the guideline of the Association for Assessment and Accreditation of Laboratory Animal Care International at the Faculty of Veterinary Medicine, Hokkaido University, Japan.

**PCB treatment.** PCBs treatment of cats has been described in our previous studies [29, 30]. In brief, eight male cats (24–28 months old, 3–5 kg, *Felis catus*) were obtained from Kitayama Labs Co., Ltd. (Japan). After 2-week acclimatization, the cats were separated into two groups: control (n = 4) and treatment (n = 4). The animals were injected intraperitoneally

with corn oil and a mixture of the major 12 PCB congeners [CB18, CB28, CB70, CB77, CB99, CB101, CB118, CB138, CB153, CB180, CB187, and CB202 ($\geq$99.5% purity; AccuStandard Inc., CT, USA)] in corn oil at a dose of 0.5 mg (each congener)/kg (body weight) once in the control and treatment groups, respectively. The concentration was set to be slightly greater than that detected in the serum of pet cats, but not lethal [32]. The blood was collected by time to time from 0, 6, 24, 48, 72, and 96 h, followed by centrifugation at 1,500xg for 15 min. The serum was collected and stored at -80˚C until biochemical analyses. The clinical signs and body weight were determined daily. After the cats were anesthetized with pentobarbital and euthanized by KCl injection, weights of organs (including the liver, brain, kidney, adrenal gland, heart, lung, pancreas, spleen, testis, small intestine, and thyroid gland, as well as visceral fat) were measured. The liver sample used for gene expression studies was preserved in RNAlater (Sigma- Aldrich Co., St. Louis, MO, USA).

**BDE-209 treatment.** Thirteen-months-old male cats (*Felis catus*, 3–4 kg, n = 6) were obtained from Kitayama Labs Co., Ltd. (Japan) and fed with animal foods containing 3,917 kcal/kg of energy, including 44.5% protein, 9.7% fat, and 2.3% fiber (Nosan Co., Yokohama, Japan) and unlimited water supply. After 3-month acclimatization, the cats were divided into two groups: control (n = 3) and treatment (n = 3). The treatment group was provided oral treatment with BDE-209 ($\geq$98% purity; Sigma- Aldrich Co., St. Louis, MO, USA) at 7 mg/kg/ week in capsule, whereas the control group was provided only capsules. Clinical signs were checked by veterinary doctors, and body weights of the control and treatment cats were measured every month. The blood samples collected at the 6th, 12th, 18th, 24th, 30th, 36th, 42nd, 48th, and 54th week were subjected to centrifugation at 1,500xg for 15 min, and the supernatant serum was then collected and stored at -80˚C until further analyses. After 1-year treatment, the cats were anesthetized with pentobarbital and euthanized by KCl injection, and various organs, including the liver, brain, kidney, adrenal gland, heart, lung, pancreas, spleen, testis, small intestine, and thyroid gland as well as visceral and subcutaneous fat, were weighed. The liver sample was preserved in RNAlater for gene expression studies.

## Serum biochemical analysis

All cat sera were analyzed for the concentrations of glutamic-oxaloacetic transaminase (GOT), glutamic pyruvic transaminase (GPT), lactate dehydrogenase (LDH), high-density lipoprotein cholesterol (HDL), total cholesterol, triglycerides (TG), albumin, total protein, total bilirubin, blood urea nitrogen (BUN), and creatinine (CRE) using a conventional serum chemical analyzer (COBAS Ready, Roche Diagnostic Systems, Basel, Switzerland) and reagent strips (Spotchem panels I and II; Arkray, Kyoto, Japan).

## Measurement of steroid hormones

Previous studies have described a method for analyzing steroid hormones, including 11-deoxycorticosterone, 11-deoxycortisol, corticosterone, cortisol, 21-deoxycortisol, aldosterone, cortisone, progesterone, 17α-OH-progesterone, androstenedione, and testosterone [33, 34]. In brief, 50 mL of serum from the control and PCB-treated cats was spiked with the internal standards [*viz.*, 11-deoxycortisol-D5 (10 ppb), 17α-OH-progesterone-$^{13}$C$_3$ (10 ppb), androstenedione-$^{13}$C$_3$ (1 ppb), corticosterone-D4 (10 ppb), cortisol-D4 (100 ppb), progesterone-D9 (10 ppb), and testosterone-$^{13}$C3 (1 ppb)], followed by the addition of 1% formic acid/DDW and the methyl tert-butyl ether (MTBE, Sigma-Aldrich, USA) and centrifugation at 10,000xg for 10 min. After centrifugation, the supernatant was collected, evaporated at 60˚C under a nitrogen steam, and dissolved with 0.2 mM ammonium fluoride in 25% MeOH (Sigma-

Aldrich, USA). A liquid chromatography-mass spectrometry system (6495 Triple Quad LC/MS, Agilent Technologies) was employed to quantify the target steroid hormones.

## Gene expression analysis

Since biochemical results suggested that BDE-209 could be associated with disturbance of lipid metabolism through liver function, the total RNA was extracted from liver tissues of the control and cats treated with BDE-209 using TRI Reagent® (Sigma Life Science, USA) and cleaned up with NucleoSpin® kit (Macherey-Nagel, Germany), followed by cDNA synthesis using the ReverTra Ace® qPCR RT Master Mix with gDNA Remover (Toyobo Co., Ltd., Life Science Department, Osaka, Japan). S1 Table summarizes primer sets of genes involved in fat metabolism, including fatty acid elongase 6 (ELOVL6), stearoyl-CoA desaturase (SCD), acetyl-CoA carboxylase alpha (ACACA), and fatty acid synthase (FASN), as well as cytochrome P450 family 4 (CYP4). qRT-PCR (StepOnePlus Real-Time PCR system, Applied Biosystems, USA) was conducted using 10 μL of the PCR reaction mixture containing Fast SYBR Green Master Mix (Applied Biosystems, USA), forward and reverse primers (Thermo Fisher Scientific, Life Technologies Japan Ltd., Japan), and cDNA of each tissue. qPCR was conducted at 95˚C for 20 s followed by 40 cycles of 95˚C for 3 s and 60˚C for 30 s. Quantification of the transcripts was performed by the ΔΔCT method [35] normalized with glyceraldehyde 3-phosphate dehydrogenase (GAPDH) and beta-actin (ACTB) genes.

## Measurement of serum PCBs and PBDEs

Takaguchi et al. (2019) reported the PCB and OH-PCB levels of cat sera; their results have been referred to for correlation analysis with biochemical parameters. Extraction of PBDE from cat serum has been described previously [36]. In brief, a serum sample (50 μL) from a cat treated with BDE-209 was spiked with $^{13}C_{12}$-labeled PBDEs and $^{13}C_{12}$-labeled OH-PBDEs and denatured with hydrochloric acid. After the addition of 2-propanol, PBDEs were extracted thrice using 50% MTBE/hexane. After partitioning with KOH, the organic phase containing PBDEs and the KOH phase containing OH-PBDEs were separated. The organic phase containing PBDEs was passed through an activated silica-gel chromatography column and concentrated. PCBs and PBDEs were then identified and quantified by GC (Agilent 6890)/MS (Agilent 5973N) analysis. OH-PBDEs were back-extracted twice using MTBE/hexane from the KOH phase acidified with sulfuric acid (pH 2). The concentrated phenolic extract was cleaned up by passing through a column packed with deactivated silica-gel, and OH-PBDEs were derivatized using N-(trimethylsilyl)dimethylamine. The derivatized solution was passed through an activated silica-gel column and concentrated. OH-PBDEs were identified and quantified by GC (Agilent 6890)/high-resolution MS (Jeol JMS-800D, Japan) in the electron-ionization mode. The target compounds were measured from corresponding $^{13}C$-internal standards by the isotope dilution method [37]. One procedural blank was analyzed in each batch to detect any possible contamination from solvents and glassware. The relative recoveries of PBDEs were 70%-134%.

## Statistical analysis

A statistical analysis was performed by using JMP Pro13 (SAS Institute, USA). The results of this experiment were presented as mean ± standard deviation (SD). The body weights, organ weights, biochemical levels, steroid hormone levels, and relative gene expressions of the control and treatment groups were confirmed for normality using the Shapiro-Wilk test and tested for homogeneity of variance using the Levene's test. Wilcoxon test and student's t-test were used to determine significant differences between the control and treatment groups. In

addition, the correlations between the levels of parent compounds (PCBs and BDE209) or their hydroxylated metabolites and the serum biochemical parameters in the serum were analyzed by Spearman's correlation. *P* value < 0.05 was considered significant.

## Results

### Effects of PCB treatment

During the experimental period, the body weight changes of cats during the time course were not significantly different between the control group and PCB treatment group (Fig 1A). Table 1 summarized the absolute and relative organ weights of the PCB treatment group in comparison with those of the control group. No significant change in the absolute and relative weights of all of the selected organs was apparent, while the relative testis weight was significantly less (*P* = 0.001) in the PCB treatment group (0.6 ± 0.1) than in the control group (0.8 ± 0.0, Table 1). Concentrations of enzymes in the liver cells (*viz.*, GOT, GPT, and LDH), HDL, total cholesterol, TG, total bilirubin, BUN, and creatinine were not altered during the

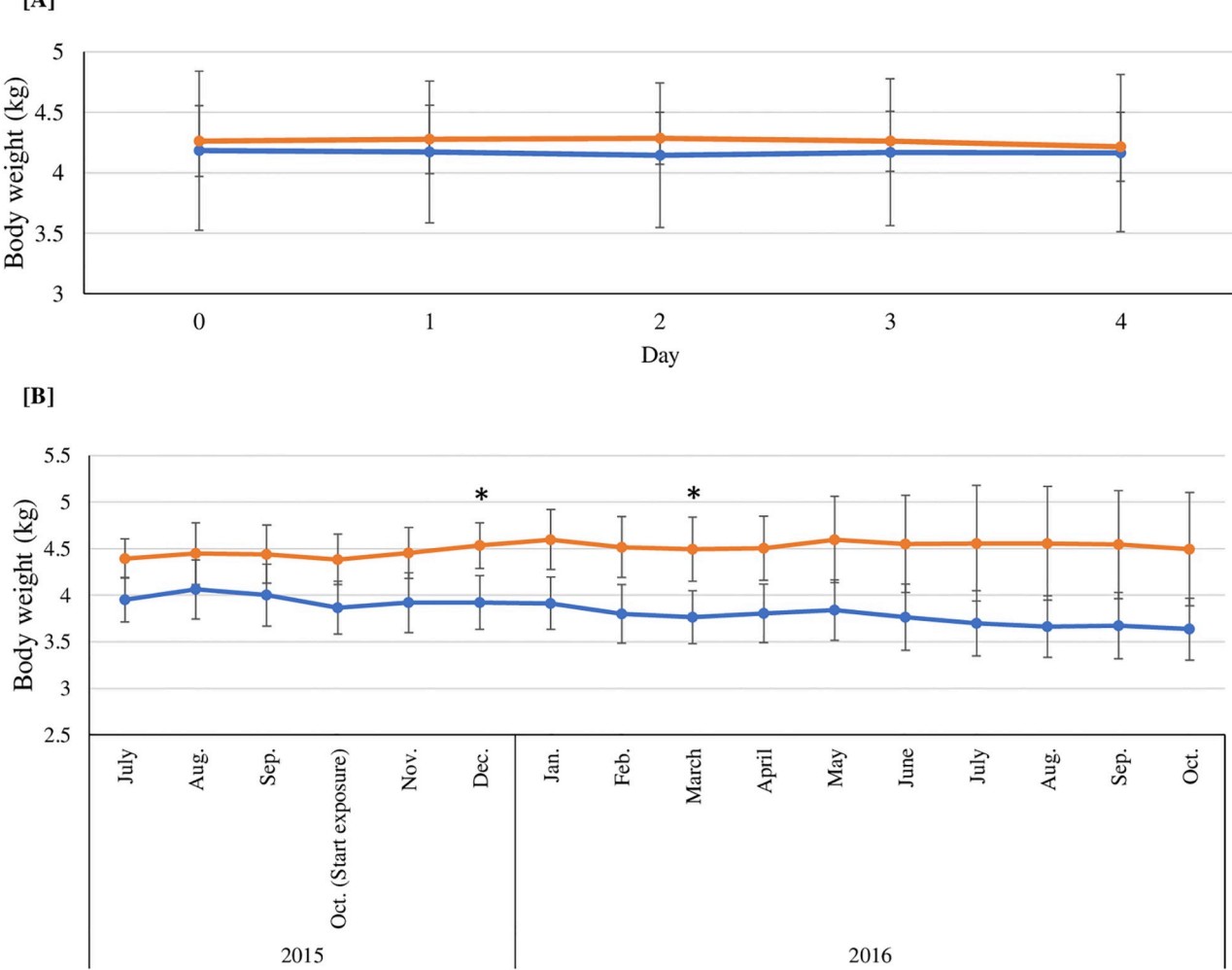

**Fig 1. Time courses of body weight changes (mean ± SD) in time courses of the control (blue) and treatment (orange) groups for PCBs [A] and BDE-209 [B] (\*: Significant differences between two groups, *P* < 0.05).**

**Table 1. Absolute and relative organ weights (mean ± SD) of the control cats and PCB-treated cats (\*: Significant differences between two groups, *P* < 0.05).**

| Organ | Absolute weight (g) | | Relative weight (g/kg bw) | |
|---|---|---|---|---|
| | Control | PCB treatment | Control | PCB treatment |
| Liver | 104.4 ± 22.8 | 115.5 ± 6.3 | 25 ± 3.0 | 27.4 ± 1.7 |
| Brain | 25.9 ± 3.7 | 24 ± 2.6 | 6.4 ± 1.5 | 5.7 ± 0.9 |
| Kidney | 44.9 ± 8.2 | 50.6 ± 6.9 | 10.8 ± 0.9 | 11.9 ± 1.3 |
| Adrenal gland | 0.6 ± 0.1 | 0.6 ± 0.2 | 0.1 ± 0.0 | 0.1 ± 0.0 |
| Heart | 13.2 ± 2.5 | 14.5 ± 0.9 | 3.2 ± 0.2 | 3.4 ± 0.2 |
| Lung | 19.8 ± 2.3 | 19.9 ± 3.2 | 4.8 ± 0.8 | 4.7 ± 0.7 |
| Pancreas | 5.4 ± 1.1 | 5.9 ± 2.3 | 1.3 ± 0.5 | 1.4 ± 0.5 |
| Spleen | 24.4 ± 11.0 | 18.3 ± 3.7 | 5.8 ± 1.9 | 4.3 ± 0.8 |
| Testis | 3.4 ± 0.7 | 2.5 ± 0.4 | 0.8 ± 0.0 | 0.6 ± 0.1\* |
| Small intestine | 86.5 ± 15.2 | 88.6 ± 9.1 | 20.8 ± 1.6 | 21 ± 2.4 |
| Thyroid gland | 0.4 ± 0.1 | 0.4 ± 0.0 | 0.1 ± 0.0 | 0.1 ± 0.0 |
| Visceral fat | 33.7 ± 14.4 | 52.5 ± 18.0 | 8.4 ± 4.0 | 12.4 ± 3.7 |

time course in the PCB treatment group in comparison with those of the control group, but the PCB treatment group exhibited a statistically significant decrease in albumin (2.9 ± 0.2, *P* = 0.02) and the total protein (6.4 ± 0.7, *P* = 0.03) levels at 96 h compared with the control group (albumin: 3.3 ± 0.2; total protein: 7.7 ± 0.6) (Fig 2). Fig 3 shows the comparison of the levels of the serum sex hormones, including 17α-OH-progesterone, androstenedione, progesterone, and testosterone in the time courses: no significant differences between both groups were observed. In PCB-treated cats, the reported levels of serum PCBs or OH-PCBs [29] were evaluated for the correlations with the levels of serum biochemical parameters. Table 2 summarizes the significant correlations (*P* < 0.05) and Spearman's rank correlation coefficients (*ρ*

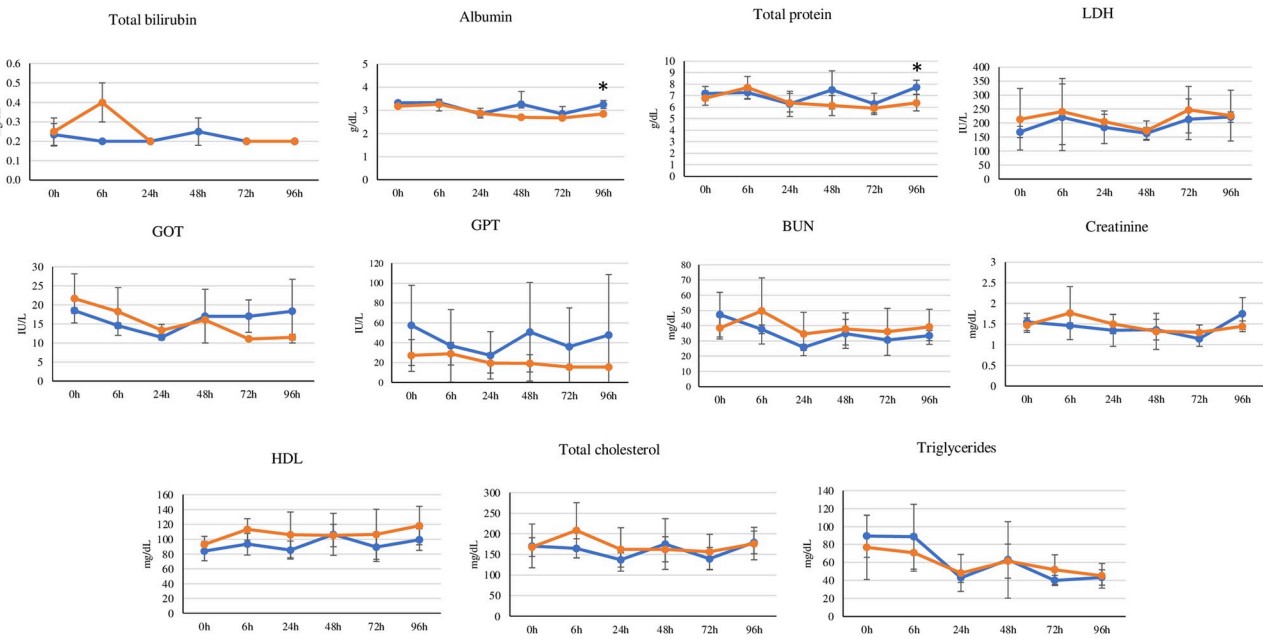

**Fig 2. Time courses of the serum biochemical changes (mean ± SD) in time courses of the control (blue) and PCB treatment (orange) groups (\*: Significant differences between two groups, *P* < 0.05).**

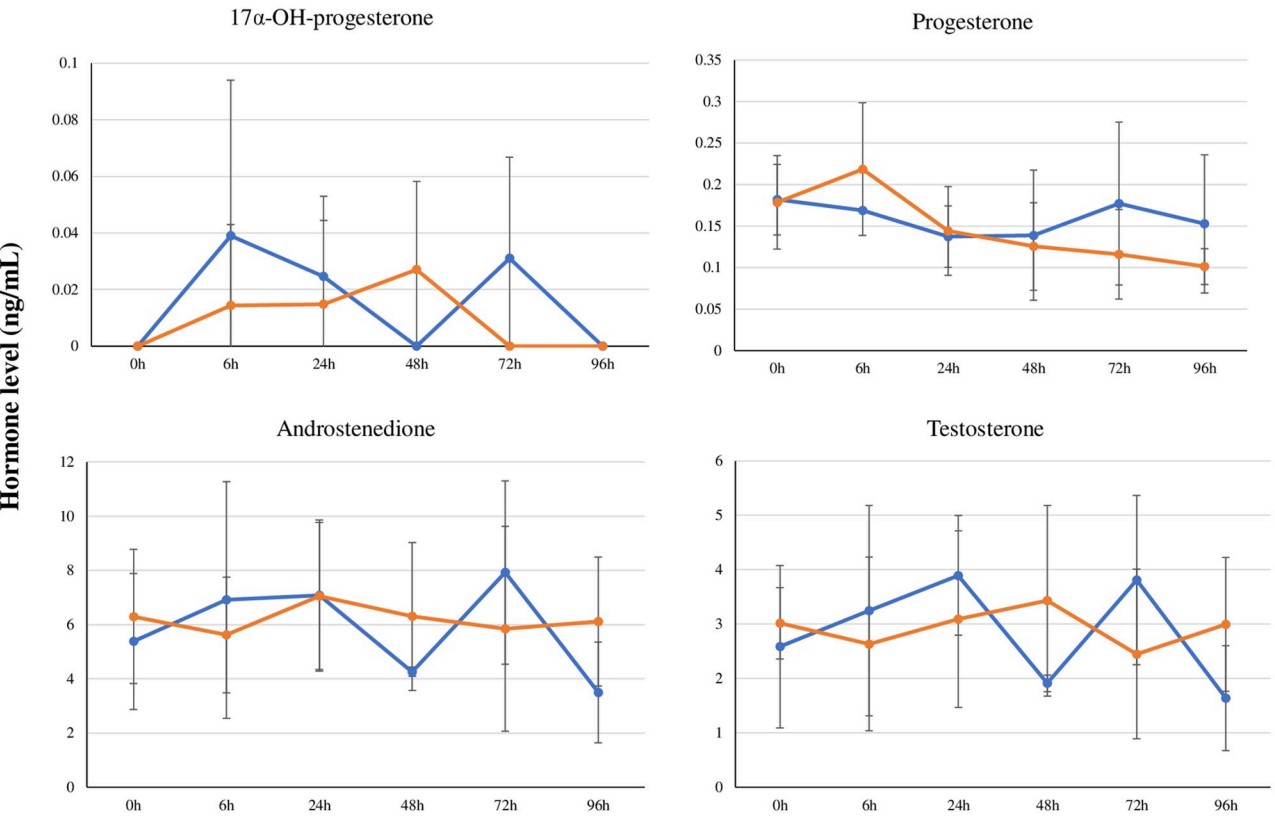

**Fig 3. Time course of the serum sex hormone levels (mean ± SD) in time courses of the control (blue) and PCB treatment (orange) groups.**

values). The levels of liver enzymes (*viz.*, GOT and GPT), TG, and progesterone were significantly negatively correlated with the parent compounds (*viz.*, CB18, CB28, and CB118) and hydroxylated metabolites (*viz.*, 4'OH-CB18, 3'OH-CB28, 4OH-CB70, 4OH-CB79, 3OH-CB101, and total OH-PCBs). Significantly positive correlations were observed between the levels of HDL and CB202 levels and the corticosterone levels and CB99, CB138, CB153, CB180, CB187, and the total PCBs, while the total protein, albumin, and creatinine were negatively related to only OH-PCBs.

### Effects of BDE-209 treatment

Fig 1B shows time courses of changes in the body weight for the control and BDE-209 treatment groups. The body weights of the BDE-209 treatment groups in the third month (4.5 ± 0.2; *P* = 0.04) and sixth month (4.5 ± 0.3; *P* = 0.04) were significantly greater than those of the control group (3.9 ± 0.3 and 3.8 ± 0.3, respectively). No significant differences in the absolute and relative weights of most organs, including the liver, kidney, adrenal gland, heart, lung, pancreas, spleen, small intestine, and thyroid gland, for the control and BDE-209 treatment groups were observed (Table 3). The relative brain weight of the treatment group (5.7 ± 0.5) was significantly less (*P* = 0.02) than that of the control group (7.4 ± 0.6), whereas the absolute and relative weights of subcutaneous fat for the BDE-209 treatment group (15.4 ± 3.9; *P* = 0.009 and 3.4 ± 0.4; *P* = 0.02) were significantly greater than those of the control group (4.8 ± 0.4 and 1.3 ± 0.2). The absolute and relative weights of visceral fat were not significantly different between the control and treatment groups, but the corresponding

**Table 2. Significant correlations (P < 0.05) and Spearman's rank correlation coefficients (ρ values) between the biochemical parameters and PCBs or OH-PCBs in the serum of PCB-treated cats.**

| Variable | by Variable | ρ | P value | Variable | by Variable | ρ | P value |
|---|---|---|---|---|---|---|---|
| GOT | CB18 | -1 | < .0001* | GOT | 4'OH-CB18 | -0.9429 | 0.0048* |
| GOT | CB28 | -0.886 | 0.0188* | GOT | 3'OH-CB28 | -0.8197 | 0.0458* |
| GPT | CB28 | -0.928 | 0.0077* | GOT | 4OH-CB70 | -0.8857 | 0.0188* |
| GPT | CB118 | -0.812 | 0.0499* | GOT | 4OH-CB79 | -0.8857 | 0.0188* |
| HDL | CB202 | 0.8857 | 0.0188* | GOT | 3OH-CB101 | -0.8697 | 0.0244* |
| Triglycerides | CB18 | -0.829 | 0.0416* | GOT | Total OH-PCBs | -0.8857 | 0.0188* |
| Triglycerides | CB28 | -0.829 | 0.0416* | GPT | 4'OH-CB18 | -0.9276 | 0.0077* |
| Corticosterone | Total PCBs | 0.8286 | 0.0416* | GPT | 3'OH-CB28 | -0.9241 | 0.0084* |
| Corticosterone | CB99 | 0.8286 | 0.0416* | GPT | 4OH-CB70 | -0.9276 | 0.0077* |
| Corticosterone | CB138 | 1 | < .0001* | GPT | 4OH-CB79 | -0.9276 | 0.0077* |
| Corticosterone | CB153 | 1 | < .0001* | GPT | 3OH-CB101 | -0.9706 | 0.0013* |
| Corticosterone | CB180 | 0.8286 | 0.0416* | GPT | Total OH-PCBs | -0.9276 | 0.0077* |
| Corticosterone | CB187 | 0.8286 | 0.0416* | Total protein | 4'OH-CB18 | -0.8407 | 0.0361* |
| Progesterone | CB28 | -0.943 | 0.0048* | Albumin | 4'OH-CB18 | -0.8857 | 0.0188* |
| Progesterone | CB118 | -0.829 | 0.0416* | Albumin | 3'OH-CB28 | -0.8804 | 0.0206* |
| | | | | Albumin | 3OH-CB101 | -0.8117 | 0.0499* |
| | | | | Creatinine | 3'OH-CB28 | -0.8804 | 0.0206* |
| | | | | Triglycerides | 4OH-CB70 | -0.8286 | 0.0416* |
| | | | | Triglycerides | 4OH-CB79 | -0.8286 | 0.0416* |
| | | | | Triglycerides | 3OH-CB101 | -0.8117 | 0.0499* |
| | | | | Triglycerides | Total OH-PCBs | -0.8286 | 0.0416* |
| | | | | Progesterone | 4'OH-CB18 | -0.8857 | 0.0188* |
| | | | | Progesterone | 3'OH-CB28 | -0.8804 | 0.0206* |
| | | | | Progesterone | 4OH-CB70 | -0.9429 | 0.0048* |
| | | | | Progesterone | 4OH-CB79 | -0.9429 | 0.0048* |
| | | | | Progesterone | 3OH-CB101 | -0.9856 | 0.0003* |
| | | | | Progesterone | Total OH-PCBs | -0.9429 | 0.0048* |

**Table 3. Absolute and relative organ weights (mean ± SD) of the control cats and cats treated with BDE-209 (*: Significant differences between two groups, P < 0.05).**

| Organ | Absolute weight (g) | | Relative weight (g/kg bw) | |
|---|---|---|---|---|
| | Control | BDE-209 treatment | Control | BDE-209 treatment |
| Liver | 88.5 ± 17.6 | 88.5 ± 16.5 | 24.3 ± 3.6 | 21.2 ± 3.5 |
| Brain | 26.9 ± 0.3 | 25.5 ± 1.3 | 7.4 ± 0.6 | 5.7 ± 0.5* |
| Kidney | 41.5 ± 5.3 | 46.1 ± 3.1 | 11.4 ± 1.3 | 10.3 ± 0.7 |
| Adrenal gland | 0.6 ± 0.1 | 0.6 ± 0.1 | 0.2 ± 0.0 | 0.1 ± 0.0 |
| Heart | 14.4 ± 2.9 | 14.3 ± 1.9 | 3.9 ± 0.5 | 3.2 ± 0.8 |
| Lung | 19.2 ± 0.5 | 20.1 ± 1.9 | 5.3 ± 0.3 | 4.6 ± 1.1 |
| Pancreas | 6.4 ± 3.8 | 6.6 ± 1.6 | 1.7 ± 0.9 | 1.5 ± 0.3 |
| Spleen | 21 ± 7.8 | 18.5 ± 9.4 | 5.8 ± 2.1 | 4 ± 1.5 |
| Testis | 2.7 ± 0.3 | 3 ± 0.3 | 0.7 ± 0.1 | 0.7 ± 0.1 |
| Small intestine | 71.1 ± 11.5 | 85.9 ± 11.0 | 19.5 ± 1.6 | 19.4 ± 3.4 |
| Thyroid gland | 0.4 + 0.1 | 0.4 ± 0.1 | 0.1 ± 0.0 | 0.1 ± 0.0 |
| Visceral fat | 85.1 ± 13.5 | 163.6 ± 62.2 | 23.8 ± 5.9 | 35.9 ± 11.8 |
| Subcutaneous fat | 4.8 ± 0.4 | 15.4 ± 3.9* | 1.3 ± 0.2 | 3.4 ± 0.4 * |

weights of the BDE-209 treatment group were greater than those of the control group. Fig 4 shows time courses of the serum biochemical changes of both groups. Statistically significant increase in the HDL levels for BDE-209-treated cats the 6th week (131.3 ± 11.6, $P$ = 0.03) and 42nd week (150.0 ± 0.0, $P$ = 0.02) and the TG level at the 48th week (50.7 ± 7.6, $P$ = 0.03) in comparison with those of the control group (97.0 ± 14.4, 113.0 ± 28.5, and 33.3 ± 5.8, respectively) was observed. Fig 5 and S6 Table show time courses of the concentrations of BDE-209 and total PBDEs in the serum of BDE-209-treated cats. At all of the time courses, the total PBDE serum levels were greater than the BDE-209 serum levels, but the trend of these levels was inconsistent. The levels of BDE-209 and total PBDEs were the highest in the 18th week (BDE-209: 701.6 ± 78.5 and total PBDEs: 889.1 ± 76.1) and the lowest in the 24th week (BDE-209: 298.1 ± 103.1 and total PBDEs: 354.9 ± 95.8). However, the detected levels of all OH-PBDEs were less. In addition, the correlations between the levels of serum biochemical parameters and total PBDEs or BDE-209 were calculated, with no significant correlation between those parameters found. As the BDE-209 group exhibited obesity, the relative mRNA expressions of the 10 genes related to lipid metabolism (*viz.*, ELOVL6, SCD, ACACA, FASN, CYP4A11, CYP4A12, CYP4B1, CYP4F22, CYP4F6, and CYP4V2) in the liver were compared between the control and BDE-209 treatment groups (Fig 6). No significant difference in the relative mRNA expression of the selected genes was observed between both groups, but the relative mRNA expressions of SCD gene were significantly lower ($P$ = 0.04 [GAPDH normalization] and $P$ = 0.02 [ACTB normalization]) in the treatment group (0.5 ± 0.1 [GAPDH normalization] and 0.6 ± 0.1 [ACTB normalization]) than in the control group (1.2 ± 0.4 [GAPDH normalization] and 1.2 ± 0.3 [ACTB normalization]).

## Discussion

To the best of our knowledge, this is the first study to provide information on the clinical signs, body weights, organ weights, serum biochemical parameters, steroid hormones, and the

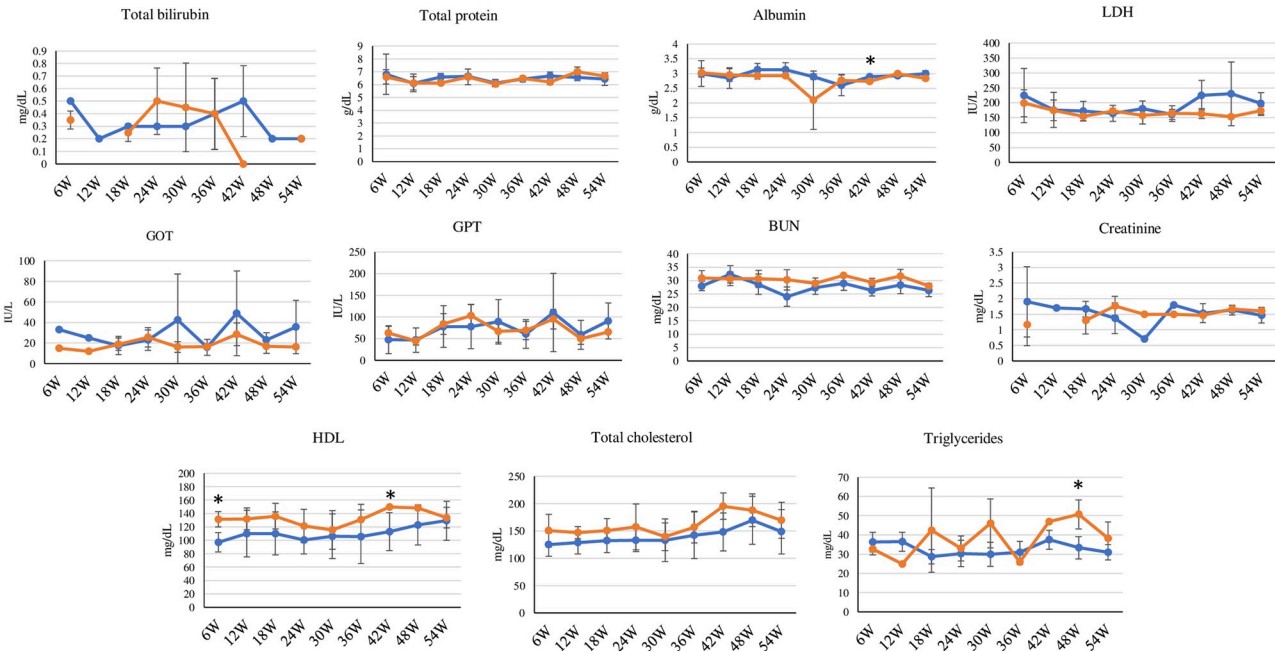

**Fig 4. Time courses of serum biochemical changes (mean ± SD) in time courses of the control (blue) and BDE-209 treatment (orange) groups (\*: Significant differences between two groups, $P$ < 0.05).**

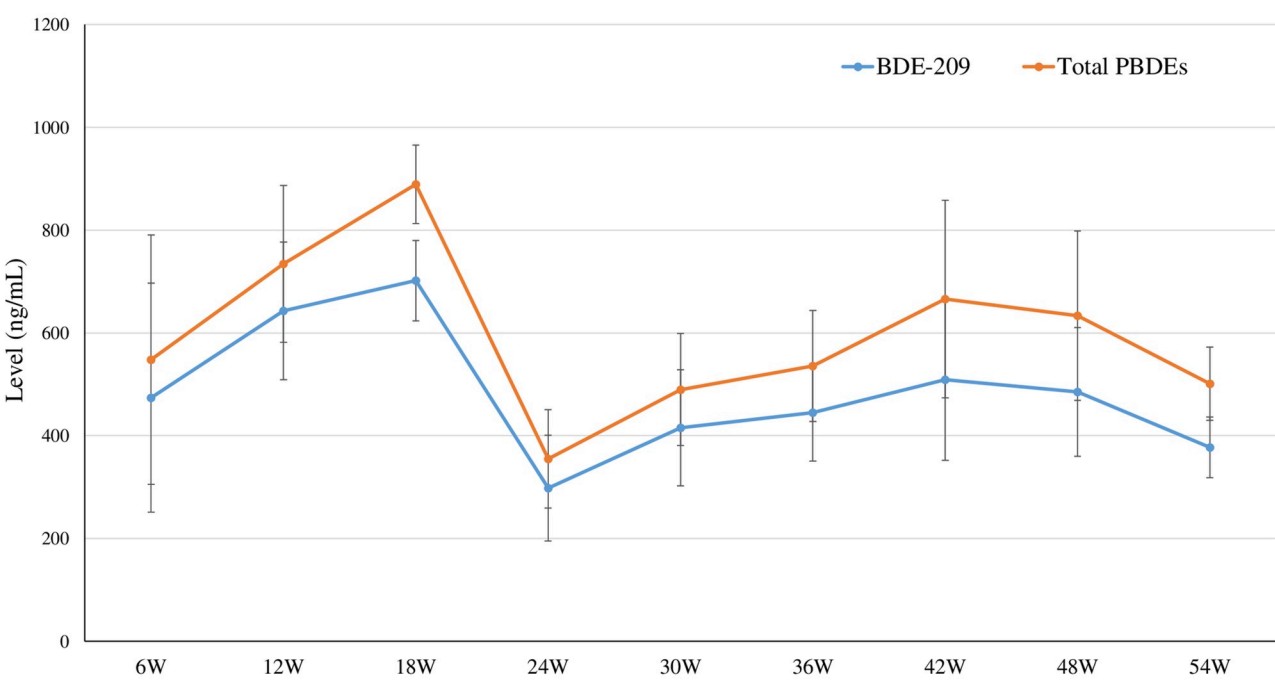

**Fig 5. Time courses of serum BDE-209 and total PBDE levels (mean ± SD) in time courses of the BDE-209 treatment group.**

gene expression related to lipid metabolism in cats subjected to short-term PCB treatment and long-term BDE-209 treatment. Among several organohalogen compounds in the household environment, PCBs and BDE209 were focused on because these chemicals are detected in the highest proportion in the serum of pet cats and humans [20–24]. Recent study on pet cats suggested that PCBs are associated with glutathione and purine metabolism and cause chronic oxidative stress, whereas PBDEs are positively associated with alanine, aspartate, and glutamate metabolism [24]. On the other hand, it has already been reported that the major OHC in Pakistani cat serum is p,p'-DDE, a DDT metabolite, with 1–2150 ng/g lipid weight, and that peta-chlorophenol (PCP) and bromophenols (BPs) are detected at relatively high levels [14], suggesting that monitoring and risk assessment of other OHCs may be necessary in the future.

In this study, cats were treated with a single dose of a mixture of 12 PCBs, which were adjusted from the administered dose of PCBs given to the dogs in a previous study [32]. The selected 12 PCB congeners, which include dioxin-like PCBs (*viz*., CB77 and CB118) and non-dioxin-like PCBs (*viz*., CB18, CB28, CB70, CB99, CB101, CB138, CB153, CB180, CB187, and CB202), have been generally found in the household environment and pet biological samples such as serum and hair [14, 17, 18, 38–41]. During the study period, the control and PCB-treated cats did not exhibit any clinical signs. In addition, no body weight changes in the time course were not observed between both groups. Similar to the investigation on dogs [32], the body weight changes and systemic toxicity were not observed in dogs (12–15 months of age, weighing 12–27 kg) treated with PCBs (5 and 25 mg/kg). These results revealed that the administered dose of PCBs (0.5 mg (each congener)/kg) could not induce changes in body weight and any clinical signs in cats.

Our results revealed that the relative testis weight of the PCB-treated group is significantly less than that in the control group. Similar to that observed for adult chickens, PCBs (50 mg/kg) did not cause significant changes in the body weight, body temperature, respiration and

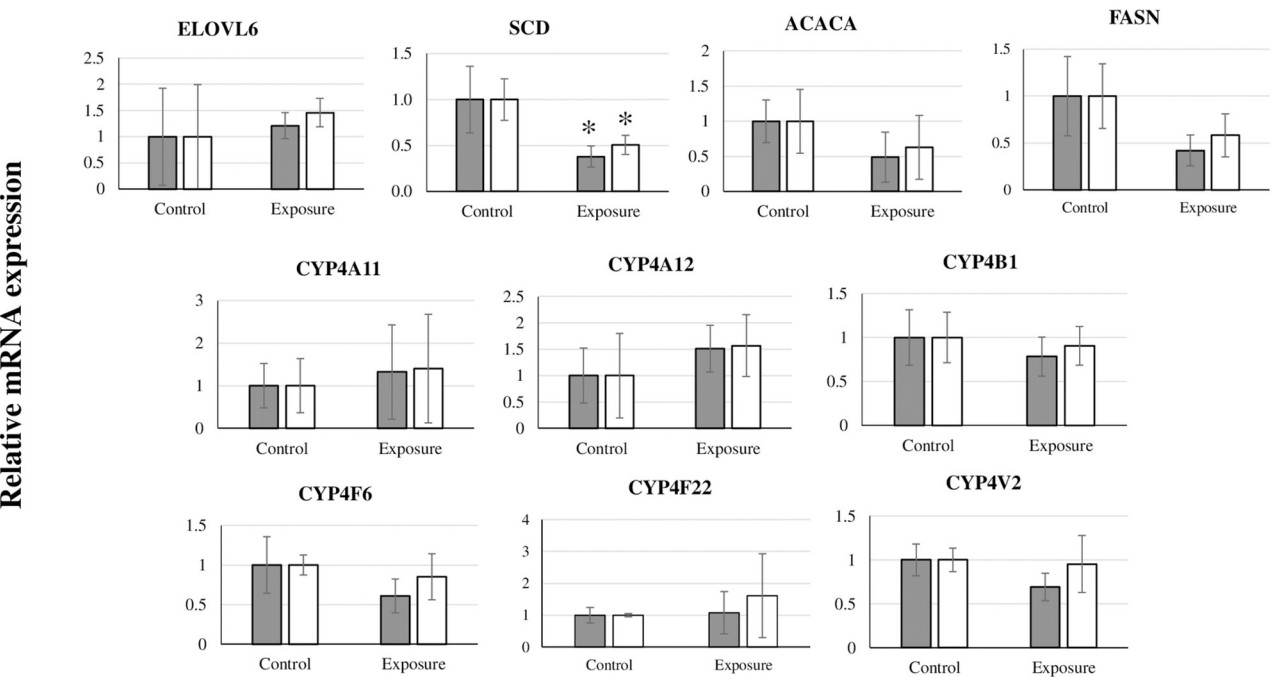

**Fig 6. Relative gene expressions (mean ± SD) in the liver of the control and BDE-209 treatment groups (reference genes: GAPDH [gray bar] and ACTB [white bar].** *: Significant differences between two groups, $P < 0.05$).

heart rates, as well as the amount of red and white blood cells, but induced a noticeable decrease in the testis weight and severely damage the seminiferous tubules [42]. These results indicated that the male reproductive system, especially the testis, could be the target organ for PCBs-induced toxicity. On the other hand, the levels of steroid hormones observed in the cat reproductive system, mainly testosterone, did not exhibit significant differences between the control and treatment groups in our study. Male adult rats treated with a single dose of PCB77 (18 and 60 mg/kg) exhibit an increase in the abnormal sperm and relative testis weight, albeit with no change in the serum testosterone concentrations [43]. This previous study and our results demonstrated that PCBs may cause various adverse effects in the testis, which is associated with the sperm production, but they may not disrupt the production of testosterone by Leydig cells.

The albumin and total protein levels of the PCB treatment group significantly decreased in comparison with those of control group, whereas the serum levels of enzymes in the liver cells, HDL, total cholesterol, TG, total bilirubin, BUN, and creatinine did not significantly change. These results revealed that the liver and kidney cells are not damaged by PCBs at the treatment level, but the balance to maintaining a stable serum albumin and protein levels (*viz.*, synthesis, distribution, or degradation) may be disturbed in cats. PCBs may interrupt albumin and protein syntheses of the liver; the hormonal balance of thyroid hormones might be a considerable factor that affects the synthesis of albumin in the liver [44]. In addition, hypoalbuminemia is characteristic of several different diseases that affect not only the liver but also the kidney and intestinal tract [45]; therefore, the excretion disturbance related to renal dysfunction should be ruled out by checking albumin and total protein levels in the urine in a subsequent investigation. At the same time, the significant negative correlations suggested that the levels of some parent congeners, including CB18, CB28 and CB118, as well as hydroxylated metabolites (*viz.*, 4'OH-CB18, 3'OH-CB28, 4OH-CB70, 4OH-CB79, 3OH-CB101, and the total OH-PCBs), may

alter the concentrations of liver enzymes (*viz.*, GOT and GPT), TG, and progesterone in cat serum. Moreover, the positive correlations indicated that the levels of CB99, CB138, CB153, CB180, CB187, and total PCBs may elevate the levels of corticosterone in PCB-treated cats, while the decrease in levels of the total protein and albumin may be affected by some OH-PCBs (*viz.*, 4'OH-CB18, 3'OH-CB28 and 3OH-CB101), which are the major metabolites in cats treated with PCBs.

Among the interesting PBDE congeners, BDE-209 is considered for use in long-term treatment because it is the major congeners found in house dust, daily products, and human serum [20, 46, 47]. Most of the available data suggested that PBDEs exhibit low short-term toxicity [48]. The treatment dose in our study was determined from the levels of BDE-209 under environmental exposure [27, 49]. Vomiting of a small amount of foods was infrequently observed in cats of the control and BDE-209 treatment groups, while abnormalities and other clinical signs were not detected throughout this study. Furthermore, the body weights of the BDE-209 treatment groups at the third month and sixth month were significantly elevated in comparison with those of the control group, suggesting that BDE-209 positively affects the body weight of male cats.

The relative weight of only the brain, among the measured organs, significantly decreased for the long-term BDE-209 treatment group. Previous studies have provided some evidence of neurotoxicity and brain developmental effects in immature mice and rats exposed to BDE-209 [50–54], but no study has reported that the brain weight decreased because of BDE-209 exposure. This result has been reported by our group for the first time: the decreased in the weight of the brain may be linked to inflammation or/and neurological diseases after BDE-209 treatment in cats. Moreover, we observed significantly increased absolute and relative weights of subcutaneous fat, and the absolute and relative weights of visceral fat for the BDE-209 treatment group tended to be higher. In contrast, previous studies in cats [15, 26] have indicated that serum PBDEs are related to feline hyperthyroidism, with typical clinical signs of weight loss [55, 56]. Our PCB treatment test has revealed that levels of thyroid hormones (*viz.*, free T4, total T4, free T3, total T3, and TSH) in the serum of cats could not be changed by PCBs treatment, as already reported previously [29].

Most of the biochemical parameters did not exhibit a significant difference in serum levels between the control and treatment groups in all of the time courses, but the serum albumin level at the 42nd week in the treatment group significantly decreased in comparison with that of the control group. These results suggested that BDE-209 at the treatment level dose not harm liver and kidney cells, but it may interfere with the synthesis of albumin in the liver or excretion in the kidney [44].

The HDL and TG levels of the BDE-209 treatment group significantly increased in comparison with those of the control group, and the total cholesterol levels tended to be higher in the BDE-209 treated group at almost all of the treatment times, but with no significantly difference between both groups. The results suggested that BDE-209 can be associated with the disturbance of lipid metabolism via liver function because the liver plays a key role in lipid metabolism [57]. Therefore, the relative mRNA expressions of genes involved in lipid metabolism in the liver were analyzed. The relative mRNA expressions of the SCD gene in the treatment group were significantly downregulated in comparison with those in the control group. SCD is a membrane-bound enzyme that is responsible for converting the saturated fatty acids to monounsaturated fatty acids during biosynthesis of fats [58, 59]. In addition, the ACACA, FAS, CYP4B1, CYP4F6, and CYP4V2 mRNA expressions tended to decrease in cats treated with BDE-209, albeit with no significance. The lipid metabolism and fatty-acid regulation mechanisms have been explained elsewhere [57, 60]. The combination of results revealed that with the increase in the subcutaneous fat weights, BDE-209 could induce lipolysis in the liver,

which is associated with secretions of TG and HDL into the blood and contribution to lipogenesis in adipose tissues such as subcutaneous fat.

Single-dose PCB treatment of cats led to the decrease in serum albumin, and total protein levels, and testis weight, while long-term BDE-209 treatment induced an increase in the accumulation of subcutaneous fat and serum TG and HDL concentrations, albeit with a decrease in the mRNA expression levels of SCD, contributing the disruption of fat accumulation of the liver. This study could reveal some aspects regarding the biological effects of PCBs and BDE-209 on cats in *in vivo* tests at concentrations relatively similar to those detected in the monitoring studies. Feline metabolic capacity is related to insufficiency of glucuronidation and lack of CYP2B; hence, cats are likely to be at potential risk under exposure to these organohalogen compounds. It is crucial to investigate in the future the detailed health effects, especially on the reproductive system, brain, and lipid metabolism, which are related to exposure to organohalogen compounds in cats.

## Supporting information

**S1 Table. Primer sets for quantitative RT-PCR analysis.**
(XLSX)

**S2 Table. Range, mean and median values of absolute and relative organ weights of the control cats and PCB treated cats (\*: $P < 0.05$).**
(XLSX)

**S3 Table. Comparison of the range, mean and median values of biochemical parameters and steroid hormones in the serum of the control cats and PCB treated cats for all time courses.**
(XLSX)

**S4 Table. Range, mean and median values of absolute and relative organ weights of the control cats and BDE-209 treated cats (\*: $P < 0.05$).**
(XLSX)

**S5 Table. Comparison of the range, mean and median values of biochemical parameters in the serum of the control cats and BDE-209 treated cats for all time courses.**
(XLSX)

**S6 Table. Serum BDE-209 and total PBDE levels (ng/mL, mean ± SD) in time courses of the BDE-209 treatment group.**
(XLSX)

## Acknowledgments

We are grateful to all members of the Laboratory of Toxicology (Graduate School of Veterinary Medicine, Hokkaido University) for taking care of cats and sincerely thank Mr. Takahiro Ichise for technical support.

## Author Contributions

**Conceptualization:** Kraisiri Khidkhan, Hazuki Mizukawa, Kei Nomiyama, Mitsuyoshi Takiguchi, Shinsuke Tanabe.

**Data curation:** Kraisiri Khidkhan, Hazuki Mizukawa, Yoshinori Ikenaka, Kei Nomiyama, Nozomu Yokoyama, Osamu Ichii.

**Funding acquisition:** Mayumi Ishizuka.

**Investigation:** Nozomu Yokoyama, Osamu Ichii, Mitsuyoshi Takiguchi.

**Methodology:** Kraisiri Khidkhan, Kei Nomiyama.

**Project administration:** Mayumi Ishizuka.

**Validation:** Hazuki Mizukawa, Yoshinori Ikenaka, Shouta M. M. Nakayama.

**Visualization:** Osamu Ichii.

**Writing – original draft:** Kraisiri Khidkhan.

**Writing – review & editing:** Hazuki Mizukawa, Yoshinori Ikenaka, Shouta M. M. Nakayama, Kei Nomiyama, Mitsuyoshi Takiguchi, Shinsuke Tanabe, Mayumi Ishizuka.

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
