## [Decision Letter · Decision Letter 0]

10 Jun 2022

PONE-D-22-09902Biological effects of polychlorinated biphenyl (PCB) and decabromodiphenyl ether (BDE-209) exposure on catsPLOS ONE

Dear Dr. Ishizuka,

Thank you for submitting your manuscript to PLOS ONE. After careful consideration, we feel that it has merit but does not fully meet PLOS ONE’s publication criteria as it currently stands. Therefore, we invite you to submit a revised version of the manuscript that addresses the points raised during the review process.

ACADEMIC EDITOR: The language of this manuscript is not very appropriate therefore author should rewrite and recheck it. Besides that few corrections are suggested by reviewer which should be incorporated in revised manuscript./>==============================

We look forward to receiving your revised manuscript.

Kind regards,

Anjani Kumar Tiwari, Ph.D.

Academic Editor

PLOS ONE

Journal Requirements:

 [This work was financially supported by Grants-in-Aid for Scientific Research (B) (KN, No. 16H02989), Challenging Exploratory Research (YI, No. 17K20038) and Scientific Research (A) (MI, No. 21H04919) from the Japan Society for the Promotion of Science and partly supported by the Ministry of Education, Culture, Sports, Science, and Technology (MEXT), Japan, to a project on Joint Usage/Research Center Leading Academia in Marine and Environmental Research (LaMer), Ehime University. The funders had no role in study design, data collection and analysis, decision to publish, or preparation of the manuscript.]

Additional Editor Comments:

Minor corrections are required before reconsideration.

Reviewers' comments:

Reviewer's Responses to Questions

**Comments to the Author**

1. Is the manuscript technically sound, and do the data support the conclusions?

Reviewer #1: Partly

2. Has the statistical analysis been performed appropriately and rigorously? 

Reviewer #1: Yes

3. Have the authors made all data underlying the findings in their manuscript fully available?

Reviewer #1: Yes

4. Is the manuscript presented in an intelligible fashion and written in standard English?

Reviewer #1: Yes

5. Review Comments to the Author

Reviewer #1: 1.As the authors claim that this is the first study to provide information on the biochemical effects of organohalogen compounds, so some more relevant studies along with risk assessment study is important. Highlight on that part.

2.Is it possible to study the biochemical effects with other organohalogen compounds (OHC) in cats (instead of PCBs and PBDEs)? Enlightenment is required.

3.Some current references (after the year: 2019) are required to the relevant field.

6. PLOS authors have the option to publish the peer review history of their article (what does this mean?). If published, this will include your full peer review and any attached files.

Reviewer #1: No

---

## [Author Response · Author response to Decision Letter 0]

29 Aug 2022

Dear Dr. Anjani Kumar Tiwari

Submission no: PONE-D-22-09902

Submission title: Biological effects of polychlorinated biphenyl (PCB) and decabromodiphenyl ether (BDE-209) exposure on cats

Thank you very much for your letter dated 10 Jun 2022 with the reviewer’s comments. 

The following are our responses to reviewer’s comments regarding this manuscript. The revised words or sentences have been highlight in yellow.

ACADEMIC EDITOR: 

The language of this manuscript is not very appropriate therefore author should rewrite and recheck it. 

This manuscript was revised and checked by an experienced editor whose first language is English.

Reviewers' comments:

Reviewer #1: 

1.As the authors claim that this is the first study to provide information on the biochemical effects of organohalogen compounds, so some more relevant studies along with risk assessment study is important. Highlight on that part.

It was revised and highlighted on the parts of the abstract, introduction, and discussion. (Page 2, Line 41-44, Page 4, Line 65-69, and Page 21-22, Line 407-409) In addition, references related to the risk assessment of OHCs are shown below;

Ma et al., 2017. Environmental Research. 2017;155:116-22. 

Mizukawa et al., Environ Sci Technol. 2016;50(1):444-52. 

Zheng et al., 2015. Environment International. 2015;78:1-7. 

Shoeib et al., 2012. Environmental Pollution. 2012;169:175-82. 

Nomiyama et al., 2022. Science of The Total Environment. 2022;842:156490. 

Walter et al., 2017. BMC Vet Res. 2017;13(1):120. 

Mensching et al., 2012. J Toxicol Environ Health A. 2012;75(4):201-12. 

Norrgran et al., 2015. Environ Sci Technol. 2015;49(8):5107-14. 

2.Is it possible to study the biochemical effects with other organohalogen compounds (OHC) in cats (instead of PCBs and PBDEs)? Enlightenment is required.

It’s possible to study with other organohalogen compounds such as DDTs, but in this study, we conducted this experiment with exposure to the PCBs and BDE-209 because there are several biomonitoring studies that detected PCBs and PBDEs have been commonly found in the blood of pet cats, and indicated health effects such as hyperthyroidism and type 2 diabetes mellitus. On the other hand, it has already been reported that the major DDT metabolite, p,p'-DDE, was the major OHC in Pakistani cat serum (N=20) and ranged between 1 and 2150 ng/g lipid weight (Ali et al., 2013, Sci Total Environ, 449:29-36), suggesting it needs to monitoring and risk assessment for other OHCs in future. This reason is highlighted in the parts of introduction (Page 3-4, Line 64-68) and discussion (Page 17, Line 301-312).

3.Some current references (after the year: 2019) are required to the relevant field.

Since the studies of PCB and PBDE residues in pet cats are limited recently (year: 2020-2022), these below references related to our study were added to this manuscript:

Mizukawa H, Nomiyama K. Biotransformation of Brominated Compounds by Pet Dogs and Cats. In: Pastorinho MR, Sousa ACA, editors. Pets as Sentinels, Forecasters and Promoters of Human Health. Cham: Springer International Publishing; 2020. p. 107-21.

Pastorinho M, Sousa ACA. Pets as Sentinels, Forecasters and Promoters of Human Health. Pets as Sentinels, Forecasters and Promoters of Human Health. 2020.

Nomiyama K, Yamamoto Y, Eguchi A, Nishikawa H, Mizukawa H, Yokoyama N, et al. Health impact assessment of pet cats caused by organohalogen contaminants by serum metabolomics and thyroid hormone analysis. Science of The Total Environment. 2022;842:156490. doi: https://doi.org/10.1016/j.scitotenv.2022.156490.

---

## [Decision Letter · Decision Letter 1]

2 Nov 2022

Biological effects related to exposure to polychlorinated biphenyl (PCB) and decabromodiphenyl ether (BDE-209) on cats

PONE-D-22-09902R1

Dear Dr. Ishizuka,

We’re pleased to inform you that your manuscript has been judged scientifically suitable for publication and will be formally accepted for publication once it meets all outstanding technical requirements.

Kind regards,

Hans-Joachim Lehmler, PhD

Academic Editor

PLOS ONE

Additional Editor Comments (optional):

Reviewers' comments:

Reviewer's Responses to Questions

**Comments to the Author**

1. If the authors have adequately addressed your comments raised in a previous round of review and you feel that this manuscript is now acceptable for publication, you may indicate that here to bypass the “Comments to the Author” section, enter your conflict of interest statement in the “Confidential to Editor” section, and submit your "Accept" recommendation.

Reviewer #1: All comments have been addressed

2. Is the manuscript technically sound, and do the data support the conclusions?

Reviewer #1: Yes

3. Has the statistical analysis been performed appropriately and rigorously? 

Reviewer #1: Yes

4. Have the authors made all data underlying the findings in their manuscript fully available?

Reviewer #1: Yes

5. Is the manuscript presented in an intelligible fashion and written in standard English?

Reviewer #1: Yes

6. Review Comments to the Author

Reviewer #1: Authors did all the work as per the comments raised in a previous round of review. Now, I think it is suitable for the publication.

Thanks

7. PLOS authors have the option to publish the peer review history of their article (what does this mean?). If published, this will include your full peer review and any attached files.

Reviewer #1: No

---

## [Editor Report · Acceptance letter]

11 Jan 2023

PONE-D-22-09902R1 

Biological effects related to exposure to polychlorinated biphenyl (PCB) and decabromodiphenyl ether (BDE-209) on cats 

Dear Dr. Ishizuka:

I'm pleased to inform you that your manuscript has been deemed suitable for publication in PLOS ONE. Congratulations! Your manuscript is now with our production department. 

Kind regards, 

on behalf of

Dr. Hans-Joachim Lehmler 

Academic Editor

PLOS ONE